suicide; India; mental health; economic development

**Corresponding author:**
Devoja Ganguli;
Email: devoja24ganguli@gmail.com

# Decriminalizing suicide: the 2017 Mental Healthcare Act and suicide mortality in India, 2001–2020

Devoja Ganguli[1] [ID], Parvati Singh[2] and Abhery Das[1]

[1]School of Public Health, University of Illinois Chicago, Chicago, IL, USA and [2]College of Public Health, The Ohio State University, Columbus, OH, USA

**Abstract**

We examine whether decriminalization of suicide in India following the 2017 Mental Health Act corresponds with changes in suicide mortality overall and by level of state development. Our study utilizes counts of suicides from the National Crime Records Bureau (NCRB) across 35 Indian states from 2001 to 2020. The exposure variable is a binary indicator for the decriminalization of suicide following 2018. We use fixed-effect Poisson regression models that include population offsets and adjust for time trends, literacy, gross state domestic product and infant mortality. We find no relation between decriminalization of suicides and overall suicide mortality (Incidence Rate Ratio (IRR): 1.037; 95% CI (0.510–2.107)). Stratification by level of state development shows that less developed states saw an increase in suicide mortality by 1.9 times following decriminalization, compared to prior years (IRR: 1.859; 95% CI (1.028–3.364)). Our findings thus indicate that decriminalization did not coincide with a decline in suicide mortality in the country, thereby highlighting the need for improved mental health infrastructure and support in India, especially in less developed states.

**Impact statement**

This study evaluates the impact of India's 2017 Mental Healthcare Act, which decriminalized suicide attempts and aimed to expand access to mental health services. Using state-level suicide mortality data from 2014 to 2021, we employed a Poisson model with state fixed effects to examine whether this legislative shift translated into reductions in suicide deaths.

Our findings show that, nationally, the policy enactment was not associated with a decline in suicide mortality. Moreover, in less developed states – characterized by lower health system capacity and limited access to mental health infrastructure – suicide rates increased in the years following the law's enactment. These findings underscore that legislative decriminalization, while important, is insufficient in the absence of accompanying investments in service delivery, mental health workforce development, and public engagement to reduce stigma.

This work has significant implications for public health and mental health policy in India and other low- and middle-income countries (LMICs) exploring similar legal reforms. It highlights the need for a systems-level approach: legal and rights-based protections must be paired with community-based mental health interventions. Without these, the intended health benefits of progressive mental health legislation may not be realized – and, in some contexts, could exacerbate existing inequities. This study provides actionable evidence that can inform the design of future mental health policy reforms and guide the strategic allocation of resources toward regions where structural constraints may inhibit the effectiveness of legal change. It also supports the case for integrating mental health planning into broader health systems strengthening efforts, particularly in underserved regions.

**Highlights**

- Decriminalization of suicide following India's 2017 Mental Healthcare Act did not lead to a significant change in overall suicide mortality across the country.
- Less developed states saw a notable increase in suicide mortality following decriminalization.
- More developed states did not show a significant change in suicide mortality.
- Decriminalization alone does not seem to have impacted suicide rates in the country overall, thereby highlighting the need for better infrastructure especially in less developed states.

## Introduction

In 2022, India reported 12.4 suicides per 100,000 population, approximately 3.9% higher than the global average (Jha, 2022). Suicides in India contribute to approximately 28% of suicides worldwide (Arya, 2024). Both the UN Sustainable Development Goals and the WHO Global Mental Health Action Plan target reducing the global suicide rate by 2030, but maintain that the criminalization of suicide in 23 countries may impede this goal (World Mental Health Report, 2022). The recent decriminalization of suicide in India may provide context for other countries attempting to reduce suicide mortality through policy measures.

In 2017, the Mental Healthcare Act restricted the application of Section 309 of the Indian Penal Code, which criminalized suicide attempts and assisted suicide through fines or imprisonment. The 2017 policy, enacted nationwide in 2018, deemed that persons attempting suicide experience severe stress and should receive care rather than criminalization. This patient-centric perspective may have salutary benefits for individuals suffering from mental illness, given the proliferation of services, as criminalization may only prevent individuals from accessing care in times of need, rather than slowing suicidal behavior. To the contrary, despite supplementing mental health care and services, policy enactment and implementation may not overcome the cultural and religious stigma associated with receiving mental health care or having the capability to expand access to care for more vulnerable groups.

Decriminalization of suicide-related self-harm and suicide attempts may also encourage accurate reporting of suicide. Estimates suggest that India may substantially underreport suicides, owing to stigmatizing legal status and penal consequences. For this reason, epidemiologic surveillance of suicide-related outcomes in India remains difficult, with some studies showing as much as 50% fewer suicides reported in formal databases relative to true incidence (Arya et al., 2021a).

Past studies on the decriminalization of suicide in other countries have found conflicting results. Studies in Ireland, Canada and New Zealand find that decriminalization does not coincide with changes in suicide deaths (Lew et al., 2022; Osman et al., 2017). Whereas an analysis of seven other nations: Canada, England, Wales, Finland, Hong Kong, Ireland and New Zealand, reports that decriminalization of suicide corresponds with an increase in suicides (Kahn and Lester, 2013).

Although suggestive, previous work on decriminalization and suicide remains limited in the following ways. First, some study samples comprise multiple countries that differ from India, given its smaller geography and population size. These studies also do not account for confounding due to differences in susceptibility by gender, economic conditions, literacy and other within-country variations. This has particular relevance given that current research on suicide trends in India shows heterogeneity within subgroups, with greater prevalence in areas of higher development, literacy and unemployment, as well as increased suicides among women (Arya et al., 2018; Dandona et al., 2018; Ramesh et al., 2022). Second, previous analyses use methodologies that may bias results, such as aggregation of suicide prevalence estimates pre- and post-legislation, as well as categorization of countries as never having criminalized suicide as opposed to not currently criminalizing suicide (Osman et al., 2017). This categorization cannot conclusively indicate that any change in suicide mortality coincides with changes in legislation (Lew et al., 2022).

We build on previous work and examine whether decriminalization of suicide in India corresponds with changes in reported suicide mortality overall and by level of state development to assess more vulnerable populations. We use data made publicly available by the National Crime Records Bureau (NCRB) to measure within-state changes in suicide over a 20-year study period in India. This examination may provide greater insight into policy measures aimed at accurately measuring and reducing suicide mortality worldwide that, in turn, may inform necessary provision and access to mental health services in targeted populations.

## Methods

### Data

We utilized suicide mortality data from the Accidental Deaths and Suicides Annual Reports created by the Indian National Crime Records Bureau (NCRB) (National Crime Records Bureau, 2024). The Government of India makes these annual reports publicly available for researchers, as well as non-governmental organizations. Used extensively in literature, they provide the most comprehensive database for suicides and crime in the country (Arya et al., 2018; Dandona et al., 2016; Patel et al., 2012; Singh et al., 2021). As our outcome, we used the count of suicides, among men and women, from 2001 to 2020 for 35 Indian states and union territories. To account for the population at risk, we obtained population denominators by state from 2001 and 2011 intercensal estimates from the Census of India (Census of India, 2024). We then conducted linear interpolation and extrapolation to calculate population estimates for 2002–2010 and 2012–2020. We used population denominators as our offset to account for changes in population over time. As with previous literature, we did not calculate age-standardized estimates of suicide mortality as NCRB does not provide age group data consistently for 2001–2020 (Arya et al., 2022). For our exposure, we used a binary indicator for the decriminalization of suicide (0, years before 2018; 1, 2018 and years after) following the enactment of the 2017 Mental Healthcare Act, which restricted criminalization in July 2018.

For covariates, we obtained the gross state domestic product per capita (GSDP), infant mortality rate (per 1,000 population) and the literacy rate (%) by state-year (Census of India, 2024). Gains in socio-economic development and modernization in India, approximated by GSDP, infant mortality and literacy, may contribute to changes in suicides (Arya et al., 2018). Infant mortality may also indicate access to healthcare and its infrastructure within states. We used indicators for the year to account for annual changes that may influence the outcome of interest. We also categorized states by socio-economic development in which less developed states belonged to the Empowered Action Group (Bihar, Chhattisgarh, Jharkhand, Madhya Pradesh, Odisha, Rajasthan, Uttar Pradesh and Uttarakhand) or among the group of Northeastern States (Arunachal Pradesh, Assam, Manipur, Meghalaya, Mizoram, Nagaland, Sikkim and Tripura). We categorized the rest of the states as more developed and excluded union territories (except for Delhi) given small sample sizes (Arya et al., 2018; Dandona et al., 2018) (Supplementary Figure A1), leaving us with 28 states for our final sample stratifying by level of state development.

### Analysis

We used a Poisson regression model with state and year fixed effects to examine within-state changes in suicide following the decriminalization of suicide in India. A fixed effects specification measured

within state changes over time and holds constant time-invariant characteristics such as culture or genetic predisposition to adverse mental health. Poisson regression models remain preferred over negative binomial when using fixed effects specifications (Allison and Waterman, 2002). We then tested suicides as a function of decriminalization by level of state development (less developed or more developed). We also stratified analyses by men and women given gender differentials found in previous literature (Dandona et al., 2018; Singh et al., 2021).

Additionally, we conducted a sensitivity test using outlier-adjusted models to assess the robustness of our findings. These models removed outliers in suicide rates below and above the 5th and 95th percentiles, respectively, to assess whether extreme values of suicide prevalence estimates drove the overall results. To test for potential misclassification of suicides, we examine whether decriminalization corresponds with greater accidental death or homicides.

## Results

Table 1 shows that suicides average 12.76 (per 100,000 population) during the entire study period. We notice an increase in prevalence in the years following the decriminalization of suicide (Figure 1). After stratifying by level of state development, more developed states have a higher average prevalence of suicide compared to less developed states (14.98 vs. 8.23 per 100,000 population) (Table 1). However, less developed states show a steeper increase in suicides following decriminalization (Figure 1).

Fixed effects Poisson regression results show that decriminalization of suicide does not coincide with changes in suicides overall in India (Incidence Rate Ratio (IRR): 1.037; 95% CI (0.510–2.107)) (Table 2). Following stratification, we find suicides increase within less developed states by 1.9 times following decriminalization, as opposed to before (IRR: 1.859; 95% CI (1.028–3.364) (Table 3). Within more developed states, we find no relation between decriminalization and changes in suicide. After adjusting for outliers, our findings remain robust to our initial inference. We continue to find an increase in suicides in less developed states following decriminalization of suicide (IRR: 1.856, 95% CI: 1.028–3.363) (Supplementary Table A1).

Further stratification by gender shows that the increase in less developed states concentrates in men (Supplementary Table A3). Supplemental analyses also show significant decreases in accidental deaths and homicides as a function of decriminalization of suicide in less developed states (Supplementary Table A4),

**Table 1.** Suicides (per 100,000 population) and sociodemographic characteristics in 35 states in India from 2001 to 2020

| Characteristics | Mean (SD) |
| --- | --- |
| Suicides (per 100,000 population) | 12.76 (10.79) |
| Suicides (per 100,000 population) – less developed states | 8.23 (8.52) |
| Suicides (per 100,000 population) – more developed states | 14.98 (6.48) |
| Infant mortality (per 1,000 live births) | 34.87 (17.33) |
| Gross state domestic product per capita (Indian rupees) | 85,694.19 (77,058.99) |
| Literacy rate (%) | 77.43 (10.01) |

which is suggestive of potential misclassification of suicides pre-decriminalization.

## Discussion

Previous work has not examined changes in suicide mortality following the decriminalization of suicide in India. Using data for 35 states in India over a 20-year period, and rigorous within-state analyses, our results show no change in overall suicide mortality as a function of decriminalization in India. Following stratification by level of state development, we find that less developed states exhibit increased suicides following decriminalization. These results add to the current literature on the global initiative to reduce suicide mortality through policy measures.

Our findings cohere with previous literature on decriminalization of suicide in other countries that show no change in overall suicide mortality (Lew et al., 2022; Osman et al., 2017). Policies such as the Mental Healthcare Act aim to rehabilitate rather than penalize suicidal individuals through mental health support. However, implementation of care may either not sufficiently address the population's needs or overcome the cultural stigma associated with mental illness. Despite the decriminalization of attempted suicide, survivors continue to be exposed to police harassment and stigma, deterring them from seeking timely medical and mental health support (Pathare et al., 2023). Our analysis showing increases in suicides in less developed states highlights that these states may not have the infrastructure necessary to enact proper mental health care through the provision of facilities or availability of psychiatrists and other medical staff. Less developed states may also not stress mental health literacy due to cultural barriers, stigma, or prioritization of other basic health-related concerns such as poverty, infectious disease, or food and water insecurity (Jenkins et al., 2013). Alternatively, we may see increases in suicides in less developed states given the elimination of fines or imprisonment due to suicide completion. Whereas we do not know if our findings indicate a true increase in suicide incidence in less-developed states, it is plausible that states that likely underreported suicides pre-2017 increased reporting following the removal of penal consequences in relation to suicide mortality (Arya et al., 2022).

Furthermore, we did not have a priori expectations of sex-specific changes in suicides post-decriminalization. Our observed increase in suicides among men after 2017 may indicate *higher* sensitivity of this group to changes in mental health policy (Arya et al., 2022). Female suicides in India present complex cultural issues stemming from misreporting of deaths following sexual assault, marriage-related abuse and exposure to gender-based violence as accidents vis-à-vis suicides (and/or homicides) (Dandona et al., 2016; Patel et al., 2021). Whereas detailed examination of cause-specific mortality among women following decriminalization of suicide in India exceeds the scope of our current analysis, we encourage future research to gauge changes in these trends post-2017 using detailed mortality data.

Experts in the field have found means restriction as an effective method for preventing suicide (Ochuku et al., 2022; Yip et al., 2012). India already implemented this tactic by banning endosulfan in 2011, a commonly available pesticide. Research shows that suicides by hanging increased in subsequent years following a decline in suicides by insecticide poisoning (Arya et al., 2021b). However, more recent research highlights that that has been a slight increase in insecticide poisoning, which hints at the possibility that people are using different pesticides to die by suicide, or buying

A. All states

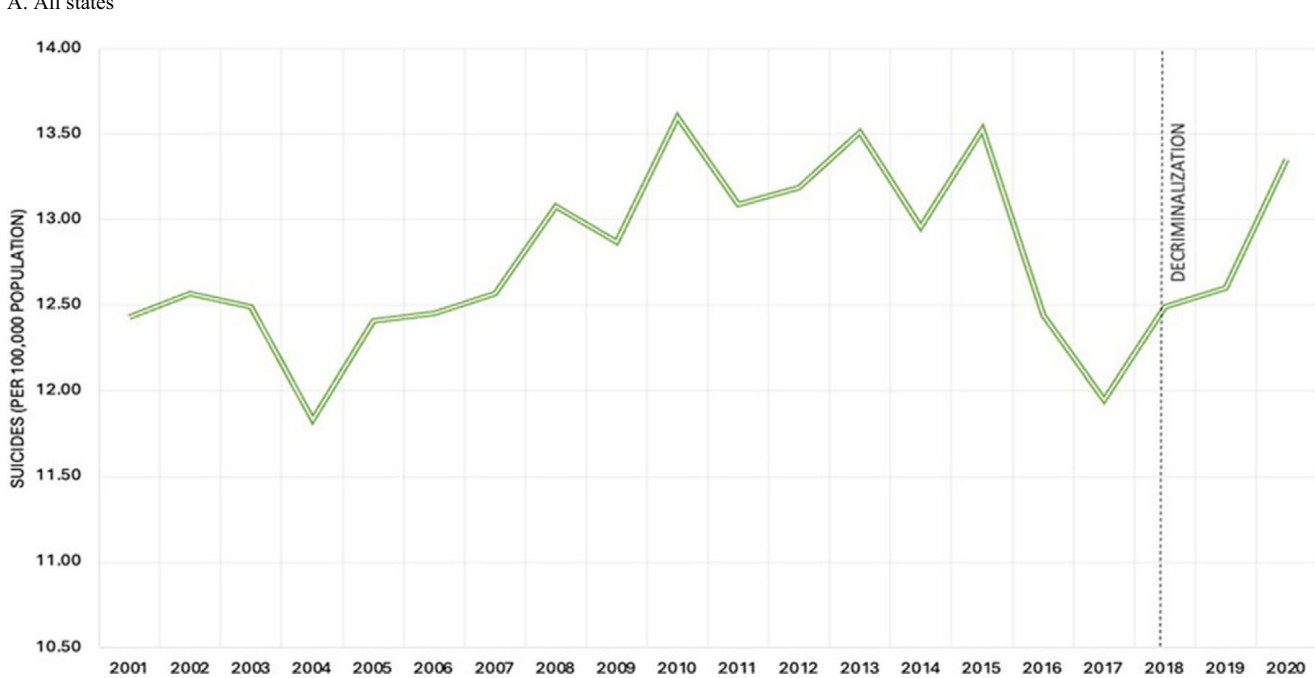

B. By level of state development

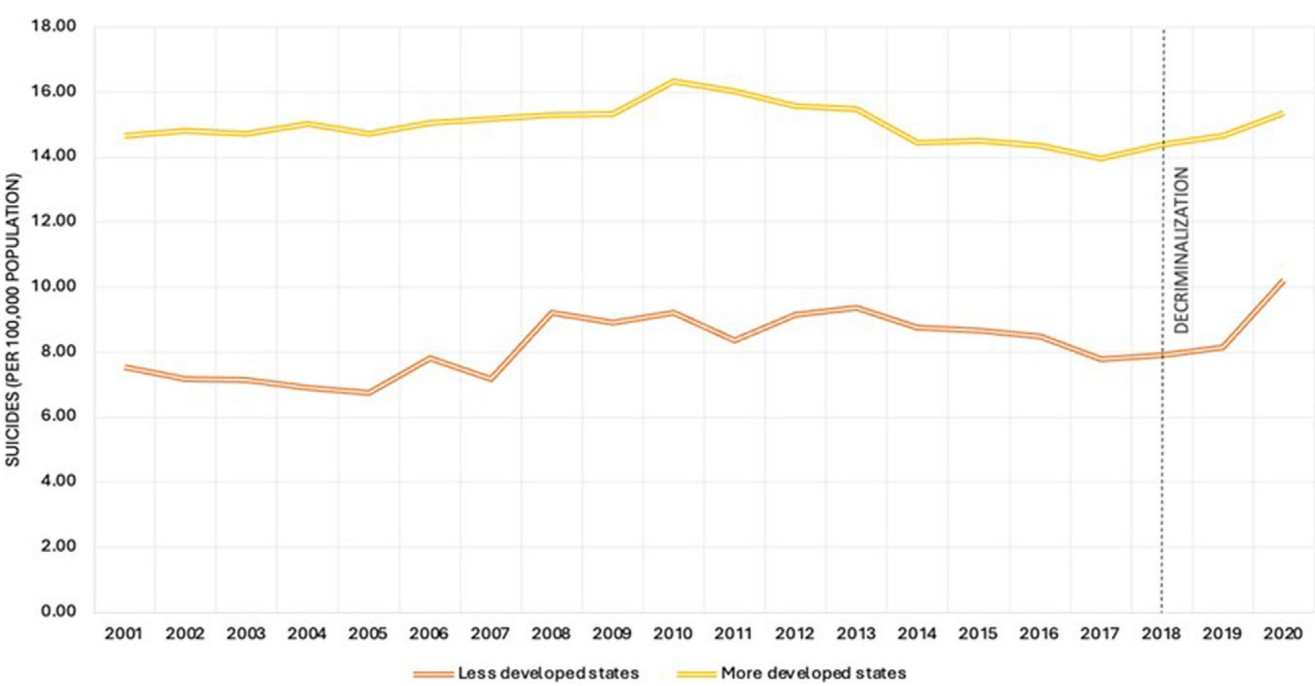

**Figure 1.** Suicides (per 100,000 population) in all 35 states in India (A) and by level of state development (B) from 2001 to 2020. The vertical line indicates the decriminalization of suicide in 2018.

endosulfan illegally (Arya et al., 2023). While Means restriction can still prove useful, the government needs to keep banning all the toxic pesticides while also concentrating on the enforcement of these bans. Additionally, policymakers should account for all mechanisms of suicide to see a notable decline in overall prevalence. Other recommendations for reducing suicide include unbiased reporting of not only suicide prevalence but also counting attempted suicides, as this will help policymakers curb suicidal ideation (Comprehensive Mental Health Action Plan 2013–2030,

2021). Encouraging private-public partnerships to increase infra-structure to accommodate the mental health needs for a country as large and diverse as India may also prove useful.

## Strengths and limitations

Strengths of our paper include the use of repeat cross-sectional data that spans over two decades and includes all 35 states in India,

**Table 2.** Fixed effects Poisson regression results predicting change in suicides (with a population offset) as a function of the decriminalization of suicide in 35 Indian states and Union Territories, 2001–2020

|  | Suicides |
| --- | --- |
| Covariates | IRR (95% CI) |
| Decriminalization of suicide (ref: before 2018) | |
| 2018 and after | 1.037 (0.510–2.107) |
| Infant mortality (per 1,000 live births) | 0.997 (0.986–1.010) |
| Gross state domestic product per capita (Indian rupees) | 1.00 (0.953–1.036) |
| Literacy rate (%) | 0.994 (0.953–1.036) |
| N (# of states) | 620 (31) |

*p < 0.1;**p < 0.05; ***p < 0.001.
N represents state-year combinations.
State and year indicators are included but not shown.

**Table 3.** Fixed effects Poisson regression results predicting change in suicides (with a population offset) as a function of the decriminalization of suicide in 28 Indian states and the Union Territory of Delhi by level of state development, 2001–2020

|  | Less developed states | More developed states |
| --- | --- | --- |
| Covariates | IRR (95% CI) | IRR (95% CI) |
| Decriminalization of suicide | | |
| Before 2018 | – | – |
| 2018 and after | 1.859 (1.028–3.364)** | 0.610 (0.211–1.762) |
| Infant mortality (per 1,000 live births) | 1.004 (0.996–1.011) | 0.999 (0.979–1.022) |
| Gross state domestic product per capita (Indian rupees) | 1.000 (0.999–1.000) | 1.000 (0.999–1.000) |
| Literacy rate (%) | 0.978 (0.946 0 1.011) | 1.003 (0.926–1.085) |
| N (# of states) | 320 (16) | 240 (12) |

*p < 0.1;**p < 0.05; ***p < 0.001.
N represents state-year combinations.
State and year indicators are included but not shown.

allowing us to observe changes in suicide prevalence over time. Compared to other studies on decriminalization, our data have the granularity to account for confounding due to socioeconomic (GSDP, literacy) and healthcare development (infant mortality). The data also allow us to test for effect modification by level of state development and gender, as found in previous work on suicide mortality in India (Dandona et al., 2018). Additionally, our fixed effects specification forces measurement of within-state changes in suicide mortality as a function of decriminalization, holding time-invariant factors constant.

This study also has limitations. Our mortality data may only comprise the lower-bound of suicide deaths in India, owing to the known rates of under-reporting in the NCRB data (Arya et al., 2021a). Our supplemental analyses show increases in accidental deaths and homicides in less developed states following decriminalization. This may indicate the absence of hydraulic relations between the reporting of suicides vis-à-vis accidental/homicidal

deaths (i.e. substitution across these causes of death), lending credence to the expectation that decriminalization may have increased reporting of suicides, particularly in regions where suicides were substantially underreported pre-2017. Previous work finds lower prevalence of suicide mortality among Muslims compared to Hindus (Arya et al., 2019; Lester, 2006)), but a higher prevalence of suicidal tendencies and attempts among Muslims residing in religiously diverse countries like India (Lester, 2006). Due to data limitations, we could not analyze suicide deaths by religion. Future research should examine how the 2017 Mental Healthcare Act affects mental health service use and suicide rates across religious groups, as these insights may prove beneficial for sister nations Pakistan and Bangladesh, as they move toward decriminalizing suicide (Mirza, 2023; Soron, 2019). Data constraints also prevented us from utilizing age-adjusted prevalence estimates of suicide mortality. When suicide data by age group becomes available, we encourage future research to utilize such estimates, as age structures may change over time within states. Age-adjusted estimates would eliminate any such bias, as each age group would contribute equally to the overall prevalence estimate (Age Adjustment – Health, United States, 2022). Finally, we do not account for the long-term lagged effects of policy implementation on suicide mortality, an avenue that warrants further investigation given that the true effects mental health policies often take time to manifest in the population.

## Conclusion

Our findings suggest that the decriminalization of suicide through the Mental Healthcare Act (MHCA) 2017 has not led to significant overall changes in suicide mortality in India, likely due to challenges in its implementation. Without adequate mental health infrastructure, awareness and access to care, policy changes alone are not sufficient to reduce incidences of suicides, especially in less developed states. Additionally, the COVID-19 pandemic has had profound effects on mental health, economic stability and social well-being, all of which could have impacts on suicidal behavior in the post-2020 era. Future research must account for these shocks when evaluating recent suicide trends. To ensure meaningful reductions in suicide mortality, policymakers may have to beyond legal reforms by strengthening mental health services, improving data transparency and addressing the social determinants of suicide, particularly in vulnerable populations.

Future research should also integrate suicide mortality estimates with national mental health data to explore how reported risk factors – such as depression, anxiety and substance use – correspond with observed suicide trends. The India National Mental Health Survey (NMHS) conducted in 2016, remains the largest mental health surveillance survey in India. NHMS aimed at assessing the prevalence, patterns and treatment gaps associated with mental health disorders across India. NMHS covered 12 states and included diverse demographic segments to offer a representative snapshot of the country's mental health landscape (Murthy, 2017; Pradeep et al., 2018, p. 201). The survey found that currently, 10.6% of the Indian population suffers from mental health disorders, while the lifetime prevalence of mental disorders in India was 13.7%. Mood disorders, anxiety, neuroses, and substance use disorders, which comprise strong risk factors for suicide, were the most common mental disorders identified. Mood disorders affected approximately 5.7% of the population, while anxiety-related disorders had a prevalence rate of 3.7%. Mental and behavioral

problems from psychoactive substance use exhibited the highest prevalence at 22.4%. Whereas the NHMS found 5% of the population at high risk of suicide, it is plausible that subsequent waves of this survey may identify a higher prevalence of suicidal ideation and self-harm following the 2017 decriminalization of suicide in India (Amudhan et al., 2020; Gautham et al., 2020; Jayasankar et al., 2022). Concerningly, the NMHS also identified a treatment gap of 84.5% for mental health across the surveyed states (Gautham et al., 2020). Understanding the links between underlying psychiatric risk factors and suicide mortality may inform the development of mental health policies and interventions, particularly in addressing gaps in care that may have widened post-decriminalization.

**Open peer review.** To view the open peer review materials for this article, please visit http://doi.org/10.1017/gmh.2025.10031.

**Supplementary material.** The supplementary material for this article can be found at http://doi.org/10.1017/gmh.2025.10031.

**Data availability statement.** The data are publicly available and can be downloaded from https://www.ncrb.gov.in/.

**Author contribution.** D.G.: Conceptualization, Methodology, Validation, Visualization, Writing – original draft, Writing – review and editing; P.S.: Validation, Writing – original draft, Writing – review and editing; A.D.: Conceptualization, Data curation, Formal analysis, Methodology, Supervision, Validation, Visualization, Writing – original draft, Writing – review and editing.

**Financial support.** None.

**Competing interests.** The authors declare that they have no known competing financial interests or personal relationships that could have appeared to influence the work reported in this paper.

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
