## [Reviewer Report]

This is a good, thorough paper worthy of publication. I especially liked how researcher conducted sensitivity analyses and tested for misclassification. Although, there is slight question over the novelty of the study given that some studies have already reported on suicide rates based on the NCRB data till 2020. Having said that, it is also true that none have looked at the impact of decriminalization of suicide on the rates of suicide, so it is justified. Please find my comments appended below

Abstract

1. Purpose: ‘We examine whether decriminalization of suicides’: suicide not suicides

2. Methods: ‘As our outcome’; ‘As our exposure’: These seem a bit awkward; try using something else like “Our study utilizes suicide counts” and “The exposure variable is a binary indicator”.

Highlights

1. ‘Decriminalization alone did not reduce suicide rates in the country, thereby highlighting the need for better infrastructure especially in less developed states’: Please use a more measured tone, given this study is based on the NCRB data which of not gold standard. Just say something like “Decriminalization of suicide does not seem to have impacted suicide rates in India” or something along those lines.

Analysis

1. ‘We used a state fixed effects Poisson regression model’: I am not sure what a “state” fixed effects model is or means.

2. I appreciate how you have conducted sensitivity analysis and tested for misclassification. Well done.

Results

1. ‘Table 1 shows that suicides average 12.76 (per 100,000 population) during the study period and increase in 2019 and 2020 after the decriminalization of suicide (Figure 1).’ Rewrite this please, it is not clear what you mean. You can break it up in two sentences if you like.

2. ‘After stratifying by level of state development, more developed states have a higher average population prevalence of suicide than less developed states (14.98 vs. 8.23 (per 100,000 population)) (Table 1).’ I don’t think you can ever have “prevalence” of suicide, people can’t “live” with suicide. Please rewrite.

3. ‘Within more developed states, we find that suicides decrease post decriminalization; however, our findings do not reach conventional levels of statistical detection (IRR: 0.610; 95% CI (0.211 – 1.762) (Table 3).’ Hmmmm, this feels a bit problematic as it reads like you want to put your point across of suicide decrease post decrim in developed states even though it’s not really true (as you yourselves mention). I would delete this and just say there was no evidence of decrease or increase.

Discussion

1. 2nd paragraph: try mentioning some of what is written in this piece: https://360info.org/how-india-continues-to-punish-those-who-attempt-suicide/

2. ‘Alternatively, we may see increases in suicides in less developed states given the elimination of fines or imprisonment due to suicide completion. Whereas we do not know if our findings indicate a true increase in suicide incidence in less-developed states, it is plausible that states that likely underreported suicides pre-2017 increased reporting following removal of penal consequences in relation to suicide mortality.’: cite this paper here: Arya, V., Page, A., Spittal, M. J., Dandona, R., Vijayakumar, L., Munasinghe, S., John, A., Gunnell, D., Pirkis, J., & Armstrong, G. (2022). Suicide in India during the first year of the COVID-19 pandemic

3. ‘Our observed increase in suicides among men after 2017 may indicate higher sensitivity of this group to changes in mental health policy’: You need to reference Arya et al 2022 here again as they have also found an increase in suicide rates among men post 2017. This is what I was on about earlier regarding the novelty of this study.

4. ‘However, research shows that suicides by hanging increased in subsequent years following a decline in suicides by insecticide poisoning (Arya, Page, Gunnell, et al., 2021).’ More recent research highlights that hanging is still going up while there a slight increase in insecticide poisoning which possibly highlights people possibly using different pesticides to die by suicide, or buying endosulfan illegally which means that the government needs to keep banning all the toxic pesticides while also concentrating of enforcement of these bans. Arya V, Page A, Vijayakumar L, Onie S, Tapp C, John A, Pirkis J, Armstrong G. Changing profile of suicide methods in India: 2014–2021. Journal of affective disorders. 2023 Nov 1;340:420-6.

5. ‘Strengths of our paper include the use of longitudinal data that spans over two decades and includes all 35 states in India.’ NCRB data is cross-sectional not longitudinal, please correct.

6. ‘Our mortality data may only comprise the lower-bound of suicide deaths in India. India considers suicides as “Medico-Legal cases” wherein hospital staff submit police reports for suicide deaths (Malathesh et al., 2022). Families may pressure hospital staff to not register suicides as “Medico-Legal cases” due to cultural stigma, resulting in NCRB undercounting suicide mortality (Ransing et al., 2022).’ Delete this and just say that suicides might be underreported in India and reference the Arya underreporting study.

7. While I am not sold on using a whole paragraph on religion and suicide (especially given the lack of evidence, which you point out), I do appreciate the inclusive and empathetic tone of the paragraph, especially the last line about sister nations and their decriminalization journey.

8. The last paragraph comes out of nowhere and starts to talk about NMHS, it almost feels like a different paper. Please delete the whole paragraph and rewrite it. Use it as a concluding paragraph where you briefly tell the reader what you did, what you found and the way forward.

---

## [Reviewer Report]

1. Title, abstract and keywords: Appropriate

2. Kindly refer to the introduction: Provide citation for “In 2022, India reported 12.4 suicides per 100,000 population, approximately 3.9% higher than the global average.”

3. Discussion:…… or food and water insecurity (Jenkins et al., n.d.). Provide the year of publication of Jenkins et al.

4. Check the reference : (Muslims India: Muslim Population in India Is Nearly 20 Crore in 2023: Govt in Lok Sabha - The Economic Times, n.d.).

5. Discussion: Kindly refer to the sentence “The survey found that 10.6% of the Indian population suffers from mental health disorders.” Mention whether the prevalence is current prevalence or life time prevalence.

6. The authors need to discuss the limitations of the study and conclusion.

7. The MHCA 2017 is not effectively implemented in the country; hence its effect is effected to be nil on suicide.

8. More so COVID-19 pandemic significantly affected the suicidal behavior post-2020. It may affect the suicide prevalence in the recent years. It need to be taken into account in the discussion.

---

## [Editor Report]

Dear Drs. Ganguli, Singh, and Das, 

I have had the chance to read the reviews of your manuscript and have myself read your manuscript. I write to invite you to revise your manuscript based on the reviewers‘ comments. Should you be able to address the reviewers’ comments I believe that your manuscript would make a strong addition to the literature and this special issue. Please don’t hesitate to reach out with any questions or concerns. 

Sincerely, 

Kristin Kosyluk, Ph.D.

Handling Editor

Special Issue: Self-harm and Suicide: A Global Priority

---

## [Editor Report]

Dear Drs. Ganguli, Singh, and Das, 

Thank you for submitting your revised manuscript. Your reponse to the reviewers' comments was thorough and they agreed that your revised manuscript should be accepted for publication in this special issue. Congratulations!

Sincerely,

Kristin Kosyluk